# Sylvatic Mosquito Viromes in the Cerrado Biome of Minas Gerais, Brazil: Discovery of New Viruses and Implications for Arbovirus Transmission

**DOI:** 10.3390/v16081276

**Published:** 2024-08-09

**Authors:** Luis Janssen Maia, Arthur Batista Silva, Cirilo Henrique de Oliveira, Fabricio Souza Campos, Leonardo Assis da Silva, Filipe Vieira Santos de Abreu, Bergmann Morais Ribeiro

**Affiliations:** 1Laboratório de Baculovírus, Departamento de Biologia Celular, Instituto de Ciências Biológicas, Universidade de Brasília (UnB), Brasília 70910-900, Brazil; luisjansm@gmail.com (L.J.M.); leocbq@gmail.com (L.A.d.S.); 2Laboratório de Bioinformática e Biotecnologia, Universidade Federal do Tocantins (UFT), Gurupi 77402-970, Brazil; arthurbatistant@gmail.com; 3Laboratório de Comportamento de Insetos, Instituto Federal do Norte de Minas Gerais (IFNMG), Salinas 39560-000, Brazil; cirilohenrique15@gmail.com; 4Programa de Pós-Graduação em Biodiversidade e Uso dos Recursos Naturais, Unimontes, Montes Claros 39401-089, Brazil; 5Centro Colaborador de Entomologia/Lacoi/IFNMG/Secretaria Municipal de Saúde de Salinas, Salinas 39560-000, Brazil; 6Instituto de Ciências Básicas da Saúde, Universidade Federal do Rio Grande do Sul (UFRGS), Porto Alegre 90035-003, Brazil

**Keywords:** sylvatic mosquitoes, RNA virome, Brazilian Cerrado, novel viral species, arbovirus transmission, biodiversity hotspot

## Abstract

Studies on animal virome have mainly concentrated on chordates and medically significant invertebrates, often overlooking sylvatic mosquitoes, constituting a major part of mosquito species diversity. Despite their potential role in arbovirus transmission, the viromes of sylvatic mosquitoes remain largely unexplored. These mosquitoes may also harbor insect-specific viruses (ISVs), affecting arboviral transmission dynamics. The Cerrado biome, known for rapid deforestation and its status as a biodiversity hotspot, offers an ideal setting for investigating mosquito viromes due to potential zoonotic spillover risks from land use changes. This study aimed to characterize the viromes of sylvatic mosquitoes collected from various locations within Minas Gerais state, Brazil. The total RNA was extracted from mosquito pools of *Psorophora albipes*, *Sabethes albiprivus*, *Sa. chloropterus*, *Psorophora ferox*, and *Coquillettidia venezuelensis* species, followed by high-throughput sequencing (HTS). Bioinformatic analysis included quality control, contig assembly, and viral detection. Sequencing data analysis revealed 11 near-complete viral genomes (new viruses are indicated with asterisks) across seven viral families and one unassigned genus. These included: *Xinmoviridae* (Ferox mosquito mononega-like virus* and Albipes mosquito Gordis-like virus*), *Phasmaviridae* (Sabethes albiprivus phasmavirus*), *Lispiviridae* (Pedras lispivirus variant MG), *Iflaviridae* (Sabethes albiprivus iflavivirus*), *Virgaviridae* (Buriti virga-like virus variant MG and Sabethes albiprivus virgavirus 1*), *Flaviviridae* (Psorophora ferox flavivirus*), *Mesoniviridae* (Alphamesonivirus cavallyense variant MG), and the genus *Negevirus* (Biggie virus variant MG virus and Coquillettidia venezuelensis negevirus*). Moreover, the presence of ISVs and potential novel arboviruses underscores the need for ongoing surveillance and control strategies to mitigate the risk of emerging infectious diseases.

## 1. Introduction

Studies on animal virome have predominantly focused on chordates and medically significant invertebrates such as ticks and mosquitoes [1]. Within mosquitoes, urban or periurban species like *Aedes aegypti*, *Culex quinquefasciatus*, and *Ae. albopictus*, known for their role in arbovirus transmission, have been the main subjects of virome investigations [2,3]. However, despite an estimated global mosquito species count exceeding 3700 [4], sylvatic mosquitoes remain underexplored regarding their viromes. This gap is significant given the critical role of certain species, such as *Sabethes* spp. in Yellow Fever Virus transmission [5] and *Coquillettidia* spp. in Eastern Equine Encephalitis virus [6] and Oropouche virus transmission [7,8].

In addition to arboviruses, mosquitoes may harbor insect-specific viruses (ISVs) which, although incapable of infecting vertebrates, offer valuable perspectives into viral evolution due to their close phylogenetic relationship with arboviruses. Some ISVs have shown the ability to modulate arboviral transmission from vectors to mammalian hosts, suggesting their potential role in developing innovative strategies for controlling arboviral outbreaks [9,10].

In parallel, high-throughput sequencing (HTS) has emerged as a critical tool for discovering new viruses, including ISVs, and determining virome in mosquitoes [11,12,13]. HTS enables the comprehensive and unbiased analysis of viral genetic material present in mosquito samples, facilitating the identification of both known and novel viruses [14]. This technology is particularly valuable for exploring the viromes of sylvatic mosquito species, which are often understudied compared to their urban counterparts [15]. By offering perspectives into the diversity and composition of mosquito viromes unavailable by other methods, HTS enhances our understanding of viral ecology and evolution, supports the development of more effective vector control and surveillance strategies, and can be utilized to discover new viruses in the largely unexplored Brazilian biomes.

The Cerrado, a biodiversity hotspot in Brazil [16], faces alarming rates of deforestation, driven in part by expanding monoculture activities [17]. Given that anthropogenic alterations in land use are major contributors to zoonotic spillover events [18], understanding the viromes associated with arthropod vectors in this biome is crucial for enhancing genomic surveillance of emerging arboviruses. Genomic surveillance methods can be used at the forefront of epidemic preparedness, as they can drive targeted surveillance of potential pathogens and studies on their biology, such as through cell cultures and animal models [18]. Therefore, the present study aims to deepen our understanding of RNA viromes within sylvatic mosquitoes inhabiting the Cerrado biome.

## 2. Materials and Methods

### 2.1. Sample Collection

Sylvatic mosquitoes were collected across five municipalities within the Minas Gerais state (Figure 1). Additionally, a mixed pool comprising diverse mosquito species was included in the study, including *Sabethes* spp. and *Psorophora ferox* (Table 1). We also analyzed *Sabethes chloropterus* and *Aedes scapularis* samples, but we did not find any viruses, so these samples were excluded from downstream analysis. The associated metadata for collection periods and land use is provided in (Appendix A).

Adult mosquitoes were captured employing the protected human attraction method (A.H.P) [19] and the Shannon trap, with assistance from entomological nets and oral aspirators [20,21]. Following capture, the mosquitoes were sorted by genera and cryopreserved in liquid nitrogen (−196 °C) before being transported to the laboratory. Taxonomic classification was conducted on a chilled table at −20 °C utilizing a stereoscopic microscope and dichotomous keys [22,23]. Mosquitoes were initially pooled per species in groups of two to 20 individuals and then grouped for HTS (Appendix A and Table 1).

### 2.2. RNA Extraction and High-Throughput Sequencing

Total RNA was extracted from mosquito pools of varying sizes (20 to 200 individuals—Table 1) using the QIAmp Viral RNA (Qiagen, Germantown, MA, USA) kit. The extracted RNA was quantified using a Quantifluor^®^ RNA system (Promega, Madison, WI, USA) kit, following the manufacturer’s instructions. For samples containing at least 1 µg of RNA, 0.1 *v/v* of 3M sodium acetate and 2 *v/v* of absolute ethanol were added. These samples were then dispatched to Macrogen (Seoul, Republic of Korea), where Illumina TruSeq stranded Total RNA library + Ribo-zero Gold libraries were generated for NovaSeq 6000, 100 bp paired-end sequencing, resulting in 20 M to 30 M reads per library.

### 2.3. Bioinformatic Quality Control, Assembly of Contigs, and Viral Detection

We conducted quality control analysis of the raw reads using fastQC v0.12.1 [24], Trim-galore for adapter removal v0.6.10 [25], and Trimmomatic v 0.39 [26] to trim 10 bp from each end of the reads. The assembly of trimmed reads into contigs was performed using Spades v3.15.5, with the -RNAviral flag [27], and only contigs with mean cover equal to or higher than 20 were further analyzed. Viral contigs were identified through local diamond blastx searches [28] of RNA-dependent RNA-polymerases compiled in RdRP-scan [29]. To identify M and S segments of *Phasmaviridae*, local diamond blastx searches were conducted using Refseq sequences deposited in Genbank [30] (https://www.ncbi.nlm.nih.gov/genbank/ assessed on 8 July 2023). Viral hits were confirmed with BLASTx searches against the non-redundant protein sequences in NCBI, and viral completeness was assessed using CheckV [31]. Viral genomes were annotated utilizing Geneious v. 11.1.5 (Biomatters) and ORFfinder (https://www.ncbi.nlm.nih.gov/orffinder/ accessed on 3 February 2024), genome figures were generated with DNA features viewer (https://edinburgh-genome-foundry.github.io/DnaFeaturesViewer/ accessed on 15 April 2024). The obtained viral sequences were deposited in Genbank with accession numbers PP946236-PP946246.

### 2.4. Phylogenetic Analyses

Global multiple alignments were created for each viral family and protein marker (protein sequences for RdRP for *Xinmoviridae*, *Phasmaviridae*, and *Lispiviridae*; full polyprotein for *Iflaviridae*, *Virgaviridae*, and *Flaviviridae*; non-structural polyprotein for *Negevirus* and ORF1a for *Mesoniviridae*) using Mafft v7.520 [32]. The resulting alignments were subsequently trimmed by employing TrimAI v1.4.15 [33] with the -automated1 flag. The amino acid substitution models for each alignment (LG + I + G4) were determined using modeltest-NG v0.1.7 [34]. Maximum Likelihood trees were then constructed utilizing RAxML-NG v1.2.0 [35] with 1000 bootstraps. All trees, otherwise stated, were rooted with an outgroup.

## 3. Results

Each sequencing library comprised approximately 20.5 to 30.4 million reads. Post-assembly, each library produced around 2014 to 37,238 contigs (Table 2). The higher contig count in the mixed mosquito pool is attributed to its diverse species composition, compared to the single-species composition of the other insect pools.

The HTS data analysis uncovered 11 near-complete viral genomes spanning seven distinct viral families and one genus of an unassigned family. Among these were the ssRNA- families *Xinmoviridae*, *Phasmaviridae*, and *Lispiviridae*, and the ssRNA+ families *Iflaviridae*, *Virgaviridae*, *Flaviviridae*, *Mesoniviridae* along with the genus *Negevirus* (Table 2). Of the 11 viral genomes described below, 7 are new, while the 4 previously known genomes were identified in new host species. The 11 contigs presented genome sizes and high completeness scores when compared to genomes of each respective taxa, as per CheckV analysis (Appendix A).

### 3.1. Xinmoviridae

We identified two consensus putative viral sequences, measuring 13 kb and 12 kb, in *Ps. ferox* and *Ps. albipes*, respectively. Their encoded RdRps exhibited 59.02% and 51.10% similarity with their closest matches in Genbank (Guadeloupe mosquito mononega-like virus, MN053735, and Gordis virus, MW435014) within the *Xinmoviridae* family (Figure 2A,B). According to ICTV criteria [36], RdRp sequences showing less than 60% similarity within this family may indicate the presence of novel genera. Both of these present genes for a putative nucleoprotein, an envelope glycoprotein, and an RdRp. Despite these viruses being discovered [37,38] in similar hosts (same Culicidae family, but different genera), their positions within the RdRp tree are notably distinct (Figure 2C). Hence, we tentatively named these viruses Ferox mosquito mononega-like virus and Albipes mosquito Gordis-like virus, respectively.

### 3.2. Phasmaviridae

Our investigation revealed three distinct putative viral segments—small (S), medium (M), and large (L) segments—in the *Sa. albiprivus* pool, with lengths of 6.5 kb, 2.1 kb, and 1.7 kb, coding for its RdRp, envelope glycoprotein and capsid protein, respectively (Figure 3A). Comparison with Genbank sequences identified an RdRp sequence showing 45.50% similarity to Miglotas virus (QRW41774.1) [38] within the *Phasmaviridae* family. As this RdRp shares less than 95% identity, as per ICTV criteria, we suggest the classification of a novel virus species, tentatively named Sabethes albiprivus phasmavirus. Phylogenetic analysis supports the distinction of this new virus, forming a separate monophyletic branch (Figure 3B).

### 3.3. Lispiviridae

We identified a consensus putative viral genome of 12 kb for the *Sa. albiprivus* pool, containing five open reading frames (ORF) (Figure 4A). Its genes include a nucleoprotein, a phosphoprotein, an envelope glycoprotein, and an RdRp. Notably, its RdRp sequence shares more than 85% similarity with that of Pedras lispivirus [39], indicating classification within the same species within the *Lispiviridae* family, according to ICTV criteria [40]. Hence, we designate this virus as Pedras lispivirus variant MG. In their study [39], previously identified this virus in *Sa. quasicyaneus*, inhabiting a transitional zone between the Cerrado and Amazon biomes. Despite geographical separation, the hosts of this virus share a close evolutionarily relationship (same subgenus—*Sabethes Sabethes*) (Figure 4B), suggesting potential occurrence in other species within this mosquito genus, a hypothesis warranting further investigation.

### 3.4. Iflaviridae

We identified a consensus putative viral genome of approximately 9 kb for the *Sa. albiprivus* pool (Figure 5A). This genome encodes a single polyprotein comprising both structural and non-structural components, showing homology to viruses from the *Iflaviridae* family. The structural component, responsible for coding the capsid proteins, has its closely related sequence in Genbank with 62% coverage and 71.65% similarity to Aedes Iflavi-like virus 1 (QQD36915.1), a virus found in *Ae. aegypti* in the Brazilian Amazon [41]. According to the criteria set by the ICTV [42], the structural component’s similarity is lower than 90%, and its occurrence in *Sa. albiprivus*, suggests that this is a novel species. We tentatively name this species Sabethes albiprivus iflavivirus (Figure 5B).

### 3.5. Virgaviridae

We identified two putative viral consensus genomes of approximately 8.8 kb and 9.1 kb in the mixed pool (*Sa. albiprivus*, *Sa. chloropterus*, and *Ps. ferox*) and *Sa. albiprivus* pool, respectively (Figure 6A), showing homology to viruses from the *Virgaviridae* family, each with a polyprotein coding gene and a putative capsid coding gene, albeit in differing synthesis. We tentatively name these viruses, respectively, Buriti virga-like virus variant MG and Sabethes albiprivus virga virus 1. In the phylogenetic analysis, Buriti virga-like virus variant MG clusters with Buriti virga-like virus found in *Sa. chloropterus* [39] (Figure 6B). Considering the ICTV criteria for taxa classification within this family (https://ictv.global/report_9th/RNApos/Virgaviridae accessed on 5 April 2024), metadata such as the mode of transmission is required. Therefore, we are currently unable to propose a taxon for Sabethes albiprivus virgavirus 1.

### 3.6. Flaviviridae

In the *Ps. ferox* pool, we identified a consensus putative viral genome of approximately 11 kb (Figure 7A) showing homology to viruses from the *Flaviviridae* family. The genome includes the ORFs characteristic of flaviviruses, encoding the polyprotein that is cleaved into structural (C, prM, E) and non-structural (NS1, NS2A, NS2B, NS3, NS4A, NS4B, NS5) proteins. NS5 encoded RdRp that presented 55.73% identity with its most closely related sequence in GenBank, Mansonia flavivirus (LC567153) [43]. We have tentatively named this virus Psorophora freaks flavivirus. Considering the ICTV criteria for the family (https://ictv.global/report_9th/RNApos/Flaviviridae accessed on 5 April 2024), classification requires metadata beyond sequence information, including antigenicity and ecological characteristics, which are beyond the scope of this study. Therefore, we lack sufficient data to propose a new taxon for Psorophora ferox flavivirus. Given its placement within the phylogenetic tree, it is most likely an ISV (Figure 7B).

### 3.7. Mesoniviridae

We identified a consensus putative viral genome consisting of 20,198 bases in *Cq. venezuelensis* (Figure 8A), showing homology to viruses within the *Mesoniviridae* family. Its gene synteny is the same as observed in the species, including genes encoding for two polyproteins for its non-structural proteins and smaller structural coding genes near its 3′ end. The encoded putative RdRp has over 99% similarity to that of Alphamesonivirus cavallyense (AXL48235.1) found in *Culex pipiens* [44]. Notably, *Cq. venezuelensis* and *Cx. pipiens* differ significantly in terms of phylogeny, behavior, and geographic distribution. These species belong to different tribes, Mansonini and Culicini, respectively. *Cq. venezuelensis* exhibits wild habits and is restricted to the Americas [45], whereas *Cx. pipiens* is urban and invasive across multiple continents [46]. In the phylogenetic analysis, this virus was found in *Cq. venezuelensis* formed a monophyletic branch (Figure 8B). We tentatively name this Alphamesonivirus cavallyense variant MG.

### 3.8. Negevirus

We identified two putative consensus viral genomes of 9.2 Kb in *Cq. velezuelensis* (Figure 9A,B), showing homology to viruses within the Negevirus genus. Both of these present a gene synteny typical of the genus, including a non-structural polyprotein, an envelope glycoprotein, and a capsid protein. One of these genomes encodes a replicase with 74% coverage and over 99% similarity to its closest sequence in Genbank, Biggie virus Mos11 (KX924639.1) found in *Culex pipiens* collected in the USA [47]. We have tentatively named this Biggie virus variant MG. As previously mentioned, these hosts differ significantly, which may suggest considerable plasticity of this virus. The second genome encodes a replicase with 73% coverage and 66% similarity to its closely related sequence, Aqua Salud Negevirus (MT197494.1) detected in *Culex declarator* in Panama [48], also a quite different host. We tentatively named this virus Coquillettidia velezuelensis negevirus (Figure 9C).

## 4. Discussion

In this study, we identified seven novel viruses and four previously known genomes identified in new host species, all of which are underrepresented in virological research. The discovery of new ISVs and potential arboviruses is particularly significant as it broadens our understanding of viral diversity within mosquito populations and reveals the potential ecological roles these viruses may play. Given the capacity of ISVs to interact with or impede arbovirus transmission [49], our findings underscore the necessity of further exploring these interactions. Moreover, HTS once again demonstrates its remarkable ability to discover new viruses, even in pools of mosquitoes from different species, thereby enhancing our understanding of comprehensive virome surveillance in regions undergoing rapid environmental changes, such as the Cerrado Brazilian biome. Compared to the work of Silva et al. (2023) [13], the number of full RdRp sequences detected was similar to our study. However, the viruses’ characterization by authors was partial. Interestingly, only one mosquito species, *Cq. venezuelensis*, was the same host searched, but the viruses identified were different. This highlights the diversity of sylvatic mosquitoes and underscores the need for further research. Below we discuss in detail the viruses detected and characterized here.

### 4.1. Xinmoviridae

Infections with anpheviruses (family *Xinmoviridae*; order *Mononegavirales*) seem to be relatively prevalent, evidenced by their presence in laboratory colonies and wild populations of Aedes albopictus from various geographic locations over several years [50,51]. Notably, *Wolbachia* bacteria enhance Aedes anphevirus replication, while this virus marginally inhibits DENV replication in mosquito cell lines [50]. Anpheviruses have also been identified in populations of *Culex* spp. [52] and anophevirus-like species have been found in *Amazonian anophelines*, including *Anopheles marajoara* and *An. darlingi* [53]. Here, we found two putative new *Xinmoviridae* species in two *Psorophora sp.* mosquitoes. The widespread distribution of most *Psorophora* species in the Americas [23] poses a potential threat to public health due to their capacity to transmit several arboviruses such as Yellow Fever Virus, Rocio, West Nile Virus, Eastern Equine Encephalitis and Ilheus Virus [54,55,56,57,58,59]. The capability of arboviruses to infect *Psorophora* species suggests a potential for disease transmission. The discovery of Ferox mosquito mononega-like virus and Albipes mosquito Gordis-like virus in *Ps. ferox* and *Ps. albipes*, respectively, presents new avenues for future studies aiming to understand the interactions between ISVs and other arboviruses in the context of pathogen transmission. The identification of these novel viruses characterized by their distinct positions within the RdRp phylogenetic tree and less than 60% similarity to known viruses, hints at the potential presence of new genera within the *Xinmoviridae* family, emphasizing the necessity for further research into the diversity and ecological roles of ISVs in mosquito populations.

### 4.2. Phasmaviridae

Phasmaviruses remain relatively enigmatic, with our understanding largely shaped by HTS endeavors. In our investigation, we introduce novel viral entities identified within *Sa. albiprivus*, tentatively classified as Sabethes albiprivus phasmavirus. These findings contribute to the expanding spectrum of phasmaviral diversity. Across arthropod hosts spanning Diptera, Hymenoptera, and Coleoptera orders, phasmaviruses have been sporadically detected [60]. Presently, the ICTV acknowledges seven genera within the *Phasmaviridae* family [61]. Nevertheless, our grasp of the physicochemical attributes of these entities remains rudimentary, such as virion size and molecular composition, with detailed characterizations limited to merely two of these genera [61]. The dearth of comprehensive in vitro analyses underscores the pressing need for elucidating the fundamental properties of phasmaviruses, including their constituent subunits. Additionally, exploring the socio-virological dynamics associated with these viruses represents a fertile ground for future inquiry, promising perspectives into their ecological roles and potential impacts on host populations. Thus, bridging these knowledge gaps stands as imperative for unraveling the enigmatic realm of phasmaviral biology.

### 4.3. Lispiviridae

Members of the *Lispiviridae* family have been detected in various hosts across different regions worldwide, including arthropods such as hemipterans, odonatans, hymenopterans, and orthopterans [60,62,63,64,65,66]. In this study, we identified a lispivirus (Pedras lispivirus variant MG) in the host species *Sa. albiprivus*, previously identified in *Sa. quasicyaneus* [39], suggesting a co-occurence between virus species and their hosts within the genus *Sabethes*, subgenus *Sabethes*. Members of the genus *Sabethes* are significant vectors of the wild yellow fever virus in the Americas [67]. Notably, *Sa. albiprivus* has already been found naturally infected by the amaryllic virus [5,68]. Future studies are warranted to investigate the occurrence of this virus in other species within this mosquito genus to enhance our understanding of its epidemiology and potential impact on public health.

### 4.4. Iflaviridae

Iflaviruses have been described in insects across several orders, including Lepidoptera, Hymenoptera, and Hemiptera, as well as in bee parasitic mites [69]. In bees, iflaviruses are major pathogens and are primarily associated with wing deformation [70,71]. Despite the phylogenetic distinction and geographical/behavioral separation between the host mosquitoes *Ae. aegypti* and *Sa. albiprivus*, they harbor similar Iflaviruses. This discovery suggests a potential broader host range and adaptability of iflaviruses across different mosquito species and regions. Further investigations are necessary to comprehend the ecological and epidemiological implications of these viruses, particularly regarding their impact on mosquito populations and their potential role in disease transmission.

### 4.5. Virgaviridae

The *Virgaviridae* family has emerged prominently across diverse mosquito virome studies, underscoring its prevalence and potential significance in mosquito ecology [72,73]. In our investigation, we introduce two novel viral entities, tentatively designated as Buriti virga- like virus variant MG and Sabethes albiprivus virgavirus 1, expanding the repertoire of known *Virgaviridae* members. Traditionally associated with plant hosts, the detection of virgavirus in mosquito viromes raises intriguing questions regarding their ecological roles and transmission dynamics within insect populations. While it is conceivable that certain virgavirus species have adapted to exploit mosquito hosts for replication, an alternative explanation implicates dietary factors in their presence within mosquito viromes. The multifaceted nature of mosquito viromes encompasses a broad spectrum of viruses, including ISVs, arboviruses, and even plant viruses [74]. Therefore, we cannot affirm whether these viruses are merely associated with mosquitos, as they could have ingested them in their plant diets or if they are indeed replicative within these animals. Further inquiries in this sense remain with other methodologies available for this analysis, such as cell cultures and small RNA sequencing.

### 4.6. Flaviviridae

The *Flaviviridae* family harbors a notable array of ISVs with intriguing implications for arboviral dynamics. Key among these are ISVs like Nhumirim virus [75], Cell fusing agent virus [76], and Culex flavivirus [77], which have been documented to exhibit inhibitory effects on their arboviral counterparts, albeit with variable outcomes across studies. While the majority of these findings stem from cell-line investigations, insights gleaned from mosquito-based studies are also noteworthy. In our study, we introduce Psorophora ferox flavivirus as a novel member of the *Flaviviridae* family. This discovery adds to the growing catalog of ISVs within the flavivirus lineage and underscores the diverse virome composition within mosquito populations. The inhibitory mechanisms attributed to ISV flaviviruses are postulated to stem from their close genetic relatedness to arboviruses within the family (reviewed in Carvalho and Long, 2021) [78]. However, the nuanced nature of these interactions is underscored by the variability in outcomes observed across different experimental settings. Despite the discordance among studies, the intricate interplay between flaviviruses holds promise for arboviral control strategies and enhances our comprehension of vector competence across mosquito species. Continued exploration of these viral dynamics, both in laboratory settings and within natural vector populations, is crucial for elucidating the full spectrum of ISV effects on arboviral transmission and for informing targeted interventions aimed at mitigating vector-borne disease burdens.

### 4.7. Mesoniviridae

Alphamesonivirus stands out as a ubiquitous presence across diverse mosquito species (over 34) and geographic regions, encompassing *Aedes* spp., *Culex* spp., *Anopheles* spp., *Armigeres subalbatus*, and *Cq. xanthogaster* [3]. Our study adds a significant dimension to this narrative by documenting the presence of Alphamesonivirus cavallyense variant MG for the first time in *Cq. venezuelensis*, a notable inclusion given the species’ unique ecological and geographical context. The robust adaptability of Alphamesonivirus across different host species and environments underscores its remarkable success in host colonization, warranting comprehensive investigation into the factors driving its epidemiological dynamics. Notably, the Yichang virus, a close relative within the *Mesoniviridae* family, has demonstrated both horizontal and vertical transmission among *Ae. albopictus* and *Cx. quinquefasciatus*, alongside the intriguing ability to hinder DENV-2 and ZIKV flaviviruses replication in mosquito cell lines [79]. However, elucidating similar relationships between other mesonivirids and arboviruses remains an open avenue for research. The discovery of Alphamesonivirus cavallyense variant MG in *Cq. venezuelensis*, a species with distinct phylogenetic, behavioral, and ecological attributes compared to its typical hosts like *Cx. pipiens*, offers a unique lens into the dynamics of mesonivirus-host interactions. By forming a distinct monophyletic branch within the phylogenetic tree, this novel variant underscores the genetic diversity and evolutionary adaptability of mesoniviruses across divergent mosquito taxa. Further exploration into the ecological drivers shaping the distribution and dynamics of mesoniviruses, alongside their potential implications for arboviral transmission dynamics, remains imperative. Unraveling the intricate interplay between mesonivirids and arboviruses holds promise for advancing our understanding of vector-borne disease ecology and informing targeted interventions aimed at mitigating mosquito-borne disease burdens.

### 4.8. Negevirus

Several viruses in this genus have been first isolated from mosquito cell lines exposed to filtered mosquito and phlebotomine specimens, particularly using C6/36 cell lines [80]. However, recent reports of new negeviruses have expanded to agricultural pests, including Hemiptera [81,82]. The detection of mosquito-associated negeviruses, using either cell cultures or HTS, has occurred across the globe [3,80,83], indicating wide host ranges and genetic variability. Their closest relatives include plant arboviruses, such as kitavirids [82,83]. Furthermore, diverse viruses in this genus also negatively modulate the multiplication of flaviviruses and alphaviruses in co-cultures in mosquito cell lines, possibly through superinfection exclusion [84,85]. This modulator capacity may be explored in the future for arbovirus control, which could help explain the lack of vectorial capacity of diverse mosquito genera in transmitting arboviruses. The negeviruses from the current study (named Biggie virus variant MG and Coquillettidia velezuelensis negevirus) represent a promising avenue for further exploration, particularly in isolating viruses from sylvatic mosquitoes using mosquito cell lines for further characterization.

Our study shares several limitations common to RNA virome studies. First, host attribution of the identified viruses is because these viruses might replicate in other organisms, such as plants consumed by mosquitos. This limitation is exacerbated by using whole-body mosquitoes for pools, rather than dissected parts such as salivary glands. To better determine the hosts of the viruses identified in our study, complementary approaches are necessary, such as virus culturing in mosquito cell lines and small RNA sequencing. The latter can provide sequences derived from the RNAi machinery of the cell, indicating active replication. Second, although none of the virus genomes we identified possess genes similar to host genes, we cannot exclude the possibility that these viruses might be endogenous viral elements (EVEs) [9]. Third, associating different segments into a single genome for multipartite viral genomes is very challenging without isolated viruses. We were able to associate the phasmavirus segments into a single genome because they were the only segments found for these viral taxa in their pool. However, this is only sometimes the case for different samples. Lastly, we used sequence-similarity approaches to identify RdRps in contigs. While RdRps are hallmarks enzymes of RNA viruses, this strategy cannot identify highly divergent RdRp sequences, leaving them in the realm of viral dark matter. Small RNA sequencing may aid in properly characterizing viral dark matter by associating sequence cleavage patterns with otherwise unrelated contigs to form complete viral genomes.

## 5. Conclusions

This study expands our understanding of the RNA virome of sylvatic mosquitoes from the Brazilian Cerrado biome, uncovering eleven near-complete viral genomes in seven distinct viral families and one genus of an unassigned family, including Ferox mosquito mononega-like virus* and Albipes mosquito Gordis-like virus* (*Xinmoviridae*), Sabethes albiprivus phasmavirus* (*Phasmaviridae*), Pedras lispivirus variant MG (*Lispiviridae*), Sabethes albiprivus iflavivirus* (*Iflaviridae*), Buriti virga-like virus variant MG and Sabethes albiprivus virga virus 1* (*Virgaviridae*), Psorophora ferox flavivirus* (*Flaviviridae*), Alphamesonivirus cavallyense variant MG (*Mesoniviridae*), and Biggie virus variant MG and Coquillettidia velezuelensis negevirus* (*Negevirus*), with seven being new and four previously known genomes identified in new host species. The discovery of two xinmovirids in *Ps. ferox* and *Ps. albipes* presents new avenues for future studies aimed at understanding the interactions between ISVs and other arboviruses in the context of pathogen transmission. Additionally, HTS once again demonstrates its remarkable capacity to discover new viruses, even in pools of mosquitoes from different species, thereby enhancing our understanding of comprehensive virome surveillance in regions undergoing rapid environmental changes, as such the Brazilian Cerrado biome. Moreover, the presence of ISVs and potential novel arboviruses underscores the need for ongoing surveillance and control strategies to mitigate the risk of emerging infectious diseases. Further research is needed to elucidate the ecological roles of these viruses, their potential impacts on vector competence, and their implications for human and animal health in the context of emerging infectious diseases. Continuing to explore the viromic landscape of sylvatic mosquitoes can better prepare us for and mitigate the threats posed by future viral outbreaks.

## Figures and Tables

**Figure 1 viruses-16-01276-f001:**
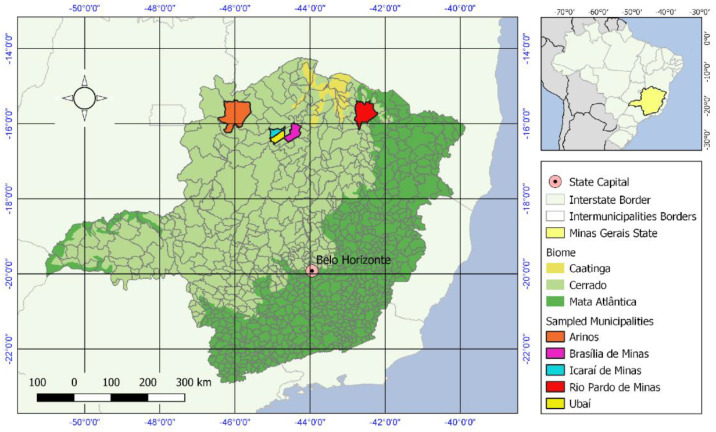
Map of Brazil and Minas Gerais state depicting the municipalities from which mosquito samples were collected and associated Brazilian biomes (Caatinga, Cerrado and Mata Atlântica).

**Figure 2 viruses-16-01276-f002:**
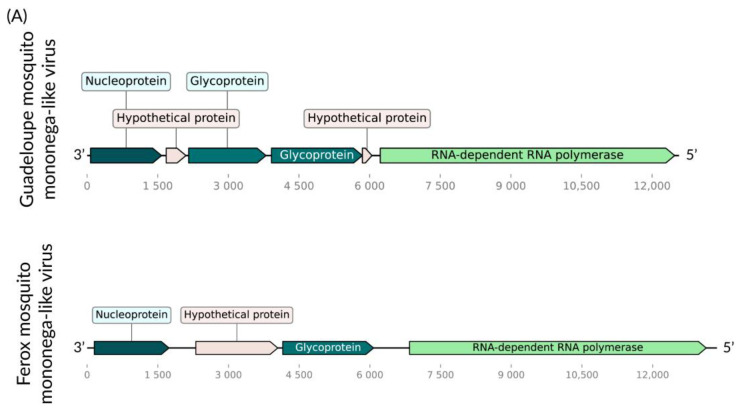
Ferox mosquito mononega-like virus and Albipes mosquito Gordis-like virus genomes discovered in *Ps. ferox* (**A**) and *Ps. albipes* (**B**), respectively, compared to their closest relatives (Guadeloupe mosquito mononega-like virus and Gordis virus, respectively). (**C**). Phylogenetic analysis of the Ferox mosquito mononega-like virus and Albipes mosquito Gordis-like virus discovered in *Ps. ferox* and *Ps. albipes* hosts, respectively in bold. ML tree was constructed using the RdRp amin acid sequences from the best hits obtained via BLASTx for each xinmovirus identified in this study against the NCBI nr-database the LG+I+G4 substitution model, scale of one aminoacid substitution per site and 1000 bootstraps. The tree includes currently recognized species in the family by ICTV, with Guadeloupe mosquito mononega-like virus Guadeloupe mosquito mononega-like virus.

**Figure 3 viruses-16-01276-f003:**
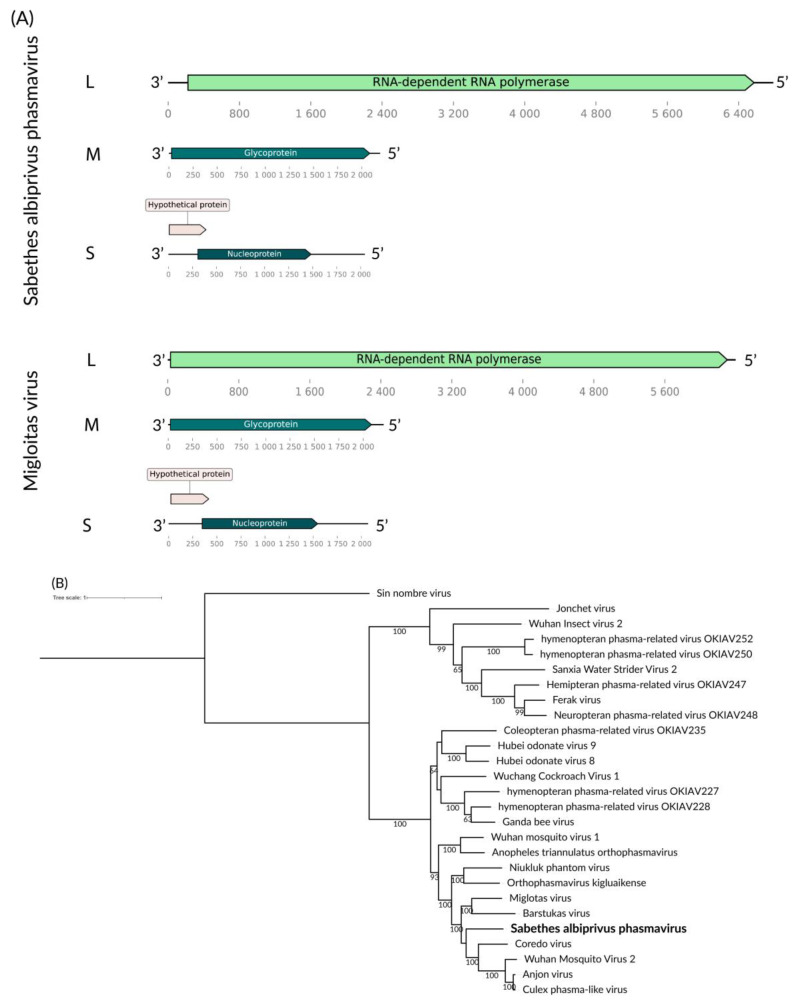
(**A**) Sabethes albiprivus phasmavirus genome exhibits L, M, and S segments identified within *Sa. Albiprivus*, shown alongside the Miglotas virus genome for reference. (**B**) Phylogenetic analysis of the Sabethes albiprivus phasmavirus discovered in *Sa. albiprivus* host in bold. ML tree was constructed using the RdRp aminoacid sequences of the best hits from BLASTx against the NCBI nr-database for the virus in the current study and the LG+I+G4 substitution model scale of one aminoacid substitution per site and 1000 bootstraps. Sin nombre virus was used as an outgroup.

**Figure 4 viruses-16-01276-f004:**
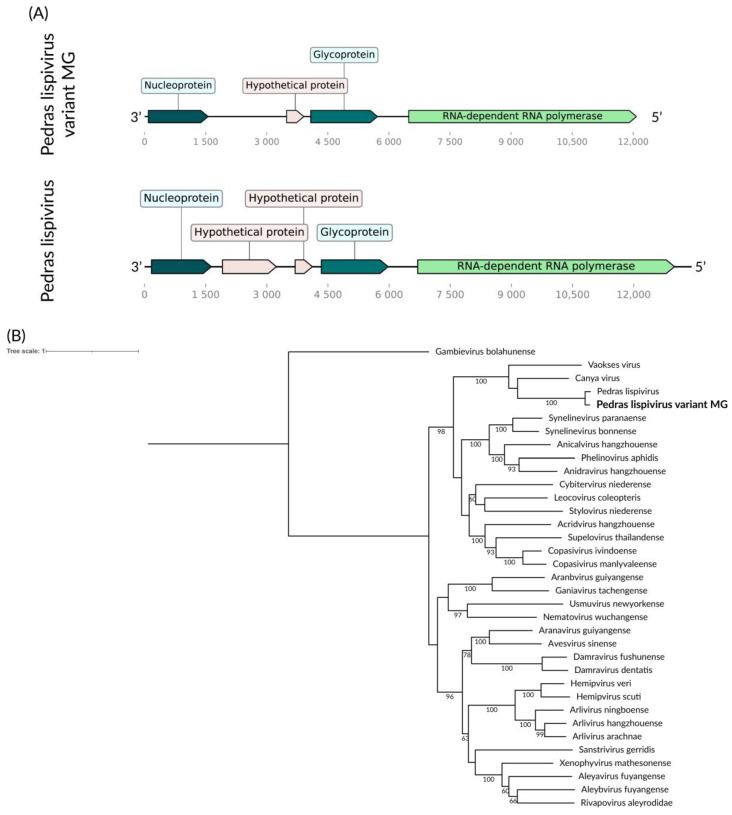
(**A**) Pedras lispivirus variant MG genome identified within *Sa. Albiprivus*, shown alongside the original variant for the species. (**B**) Phylogenetic analysis of the Pedras lispivirus variant MG discovered in the *Sa. albiprivus* host in bold. ML tree was constructed with the RdRp aminoacid sequences of the best hits found in the present study against the NCBI nr-database, currently recognized species in the family by ICTV the LG+I+G4 substitution model, 1000 bootstraps, scale of one aminoacid substituition per site and Gambievirus as an outgroup.

**Figure 5 viruses-16-01276-f005:**
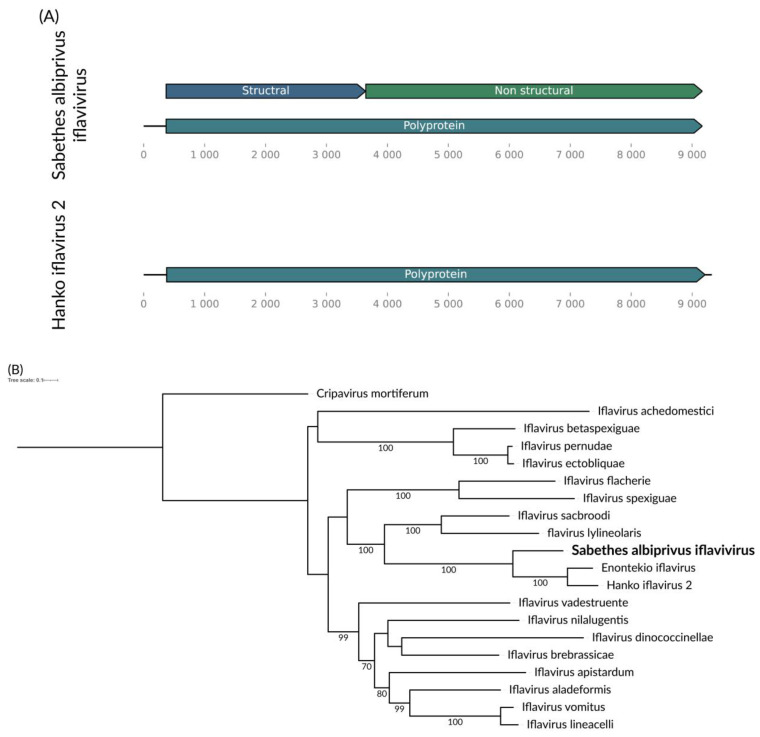
(**A**) Sabethes albiprivus iflavivirus genome identified within *Sa. Albiprivus*. (**B**) Previously described Yongsan iflavirus (Genbank accession number NC_040587.1) genome for reference. (**B**) Phylogenetic analysis of the Sabethes albiprivus iflavivirus discovered in the *Sa. albiprivus* host in bold. The ML tree was constructed using the polyprotein sequences of the best BLASTx hits found in the present study against the NCBI nr-database currently recognized species in the family by ICTV the LG+I+G4 substitution model 1000 bootstraps, the scale of 0.1 amino acid substitution per site and Cripavirus as an outgroup.

**Figure 6 viruses-16-01276-f006:**
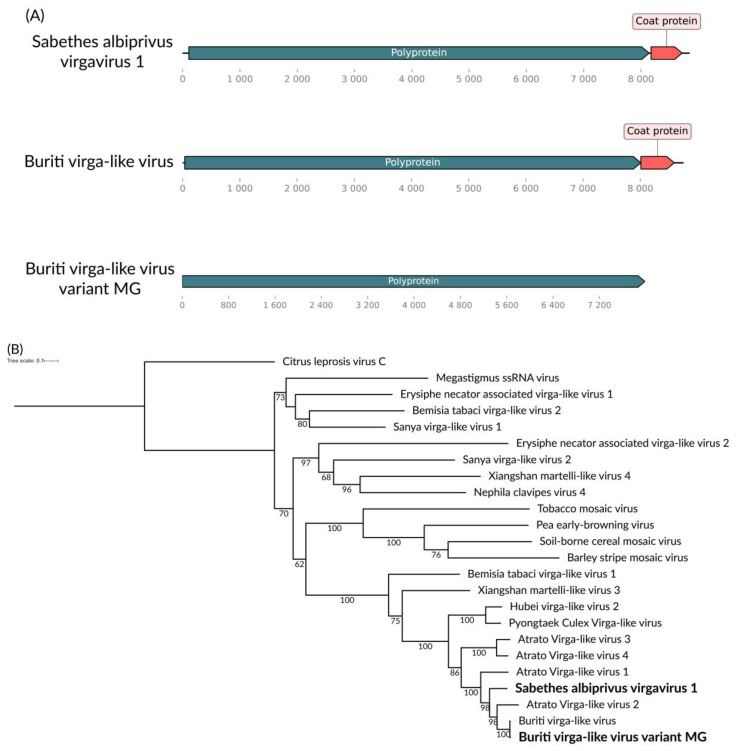
(**A**) Buriti virga-like virus variant MG (**A**) and Sabethes albiprivus virgavirus 1 (**B**) genomes identified within *Sa. albiprivus*. (**B**) Phylogenetic analysis of the Buriti virga-like virus variant MG and Sabethes albiprivus virgavirus 1 discovered in *Sa. albiprivus* host in bold. The ML tree was constructed using the polyprotein amino acid sequences of the best hits in BLASTx against the NCBI nr-database for the viruses in the current study, the LG+I+G4 substitution model, 1000 bootstraps, scale of 0.1 amino acid substitutions per site and Citrus leprosis virus as an outgroup.

**Figure 7 viruses-16-01276-f007:**
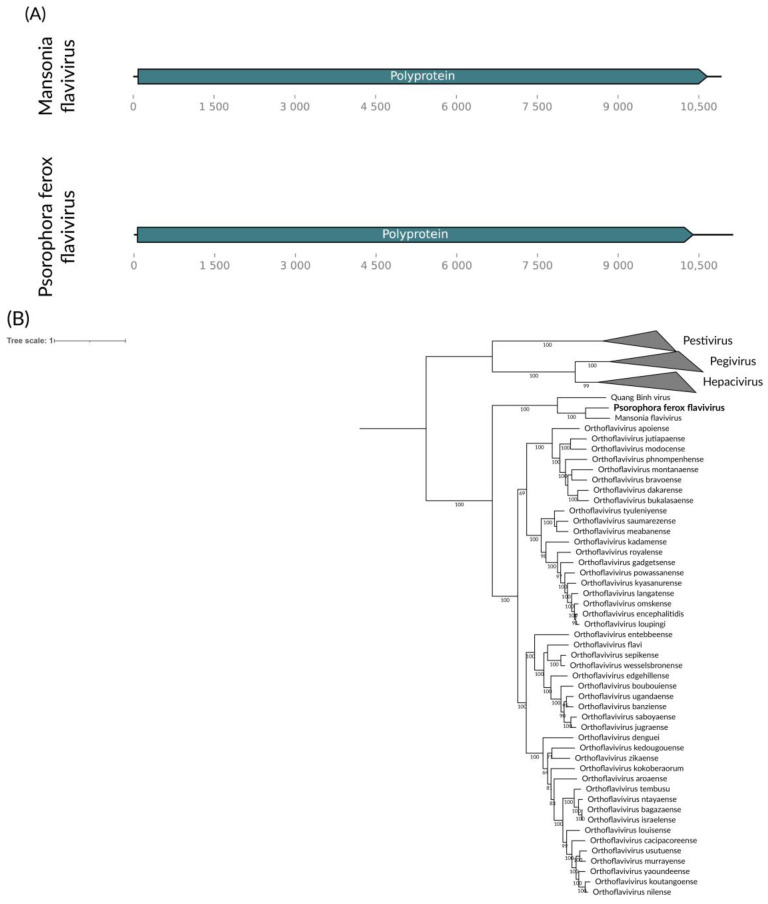
(**A**) Flavivirus genome identified within *Ps. ferox*. (**B**) Phylogenetic analysis of the Psorophora ferox flavivirus discovered in the *Ps. ferox* host in bold. The ML tree was constructed using the polyprotein sequences of the best hits from BLASTx found in the present study against the NCBI nr-database, the LG+I+G4 substitution model, 1000 bootstraps, and scale of one amino acid substitution per site. The Pestivirus, Pegivirus, and Hepacivirus genera were collapsed for better visualization.

**Figure 8 viruses-16-01276-f008:**
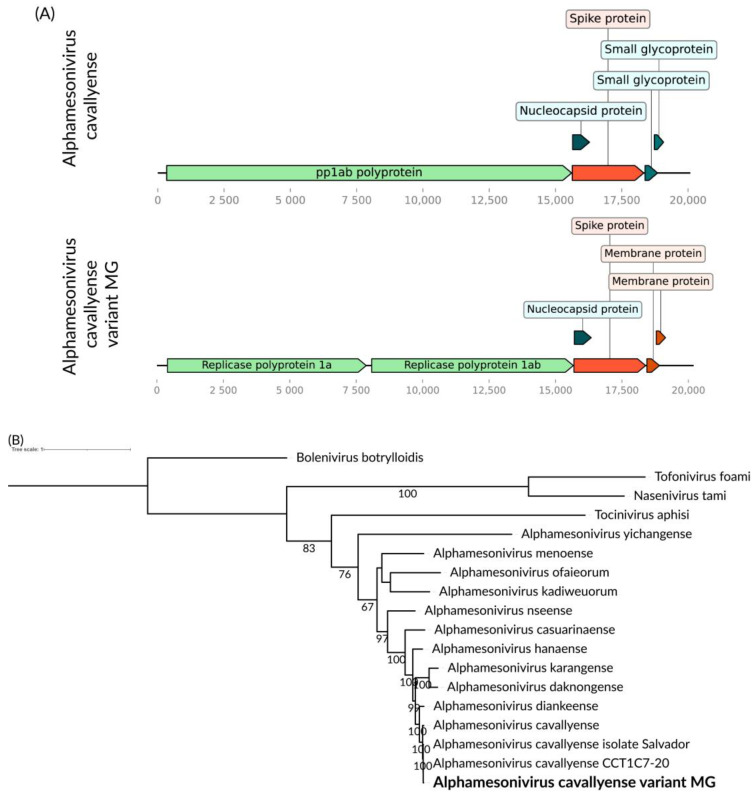
(**A**) Alphamesonivirus cavallyense variant MG genome identifies within *Cq. venezuelensis*. (**B**) Phylogenetic analysis of the Alphamesonivirus cavallyense variant MG discovered in *Cq. venezuelensis* host in bold. ML tree was constructed using the aminoacid sequences of the ORF1a replicase from representatives of the four currently recognized genera within the family, along with the two best BLASTx hits from the nr-database, the LG+I+G4 substitution model, 1000 bootstraps, scale of one aminoacid substitution per site and Bolenivirus as an outgroup.

**Figure 9 viruses-16-01276-f009:**
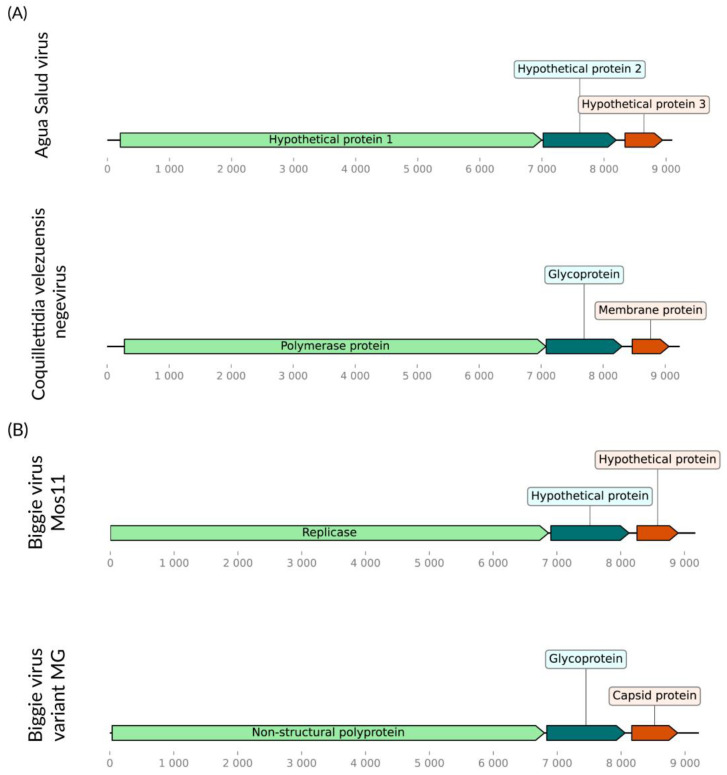
Negevirus genomes identified within *Cq. velezuelensis*. (**A**) Coquillettidia velezuelensis negevirus. (**B**) Biggie virus variant MG. (**C**) Phylogenetic analysis of the negeviruses discovered in *Cq. velezuelensis* host in bold. ML tree was constructed using the non-structural polyprotein aminoacid sequences of the best hits from BLASTx against the NCBI nr- database for the virus in the current study, the LG+I+G4 substitution model, 1000 bootstraps, scale of one aminoacid substitution per site and Citrus leprosis virus as an outgroup.

**Table 1 viruses-16-01276-t001:** Description of Species, Number of Mosquitoes, and Municipality in Minas Gerais state where mosquito samples were collected.

Municipality	Total Mosquitoes per Pool	Species per Pool
Brasília de Minas	21	*Psorophora* (*Janthinosoma*) *albipes* (Theobald, 1907)
Icaraí de Minas and Ubaí	197	*Sabethes* (*Sabethes*) *albiprivus* (Theobald, 1903)
Icaraí de Minas, Arinos, Brasília de Minas	56	Mixed pool (*Sa. albiprivus*, *Sa.* (*Sabethoides*) *chloropterus* (von Humboldt, 1819), and *Ps. ferox*)
Rio Pardo de Minas	22	*Psorophora* (*Janthinosoma*) *ferox* (von Humboldt, 1819)
Arinos	92	*Coquillettidia* (*Rhynchotaenia*) *venezuelensis* (Theobald, 1912)

**Table 2 viruses-16-01276-t002:** Viral diversity in insect pools included read numbers, contigs, viral families, genome type, and viruses found in this study including new viruses.

Viruses Found in This Study (* New Viruses)	Genome Type	Viral Families	Viral Contigs	Contig Amount	Read Number (Millions)	Insect Pool
Albipes mosquito Gordis-like virus *	ssRNA−	*Xinmoviridae*	1	3882	20.5	*Ps. albipes*
Sabethes albiprivus phasmavirus *	ssRNA− ^1^	*Phasmaviridae*	4	13,548	25.1	*Sa. albiprivus*
Pedras lispivirus variantisolate MG	ssRNA−	*Lispividae*				
Sabethes albiprivus iflavivirus *	ssRNA+	*Iflaviridae*				
Sabethes albiprivus virgavirus 1 *	ssRNA+	*Virgaviridae*				
Buriti virga-like virus variantisolate MG	ssRNA+	*Virgaviridae*	1	37,238	26.2	Mixed (*Sa. albiprivus*, *Sa. chloropterus*, and *Ps. ferox*)
Ferox mosquito mononega-like virus *	ssRNA−	*Xinmoviridae*	2	2859	30.4	*Ps. ferox*
Psorophora ferox flavivirus *	ssRNA+	*Flaviviridae*				
Alphamesonivirus cavallyense variantisolate MG	ssRNA+	*Mesoniviridae*	3	2014	25.6	*Cq. venezuelensis*
Biggie virus variantisolate MG	ssRNA+	*Negevirus*				
Coquillettidia velezuensis negevirus *						

* Asterisks indicate new viruses. ^1.^ Segmented.

## Data Availability

The authors declare that all data supporting the findings of this study are available within the paper. The viruses’ genome sequences have been deposited in GenBank under the accession numbers PP946236 to PP946246. HTS raw data has been deposited in the SRA platform under BioProject accession number PRJNA1126168, with BioSample accession numbers SAMN41926027 to SAMN41926031 and SRA (SRR) accession numbers SRR29474085 to SRR29474089.

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
