# Peer review of "Sylvatic Mosquito Viromes in the Cerrado Biome of Minas Gerais, Brazil: Discovery of New Viruses and Implications for Arbovirus Transmission"

_viruses, 2024, doi:10.3390/v16081276_

Round 1

Reviewer 1 Report

Comments and Suggestions for Authors

Abstract

Line 40 – This sentence is unnecessary. Lines 32-34 convey the same information.

Line 41 – “new viruses and potential new genera were discovered…” This phrase is redundant and should be removed or rewritten.

Introduction

Line 69 – “intricate perspectives” – what exactly does this mean?

Lines 75 -78 – This idea needs to be explained in more detail. While surveillance of arthropods for potential emerging viral threats is important, unless there is a connection between the viruses found in arthropods and disease in animals and/or humans it will be hard to garner support for continued monitoring. Isolation of live virus is also necessary to conduct the research needed to determine if the virus is a threat.

Materials and Methods

2.1 – What land use category were the collections made in? Were the mosquitoes from a single collection time? How long was the collection time? When were the collections made? Were males and females included in the pools? Were other species collected but removed from analysis?

2.3 – Viruses were classified only by similarity with known RdRPs? If you had nearly complete genomes, why not compare the whole genome? How many reads were assembled into each viral contig? How complete were the genomes?

2.4 – What is the protein marker? All of the trees shown have outgroups.

Results

The completeness of the genomes is not described, only the approximate length.

In all of the figures, the maps of the genomes are not very informative and are difficult to read. It would be beneficial for readers who aren’t familiar with these families to have the new genomes aligned with a reference sequence.

A figure should include all parts and describe all parts in a single caption.

The RdRP trees are very difficult to read. The caption needs to include the model of evolution that was used to create the tree and what the tree scale is.

By calling a virus an “isolate” it suggests that a live virus is associated with the sequence. This is not the case with these sequences.

Discussion

Line 387 – This is an awkward way of saying “we found”.

Line 392 – What is meant by physiochemical attributes?

Line 408 – The data presented here is far too limited to make this conclusion.

Line 423 – The data presented here is far too limited to make this claim.

Line 439 – How is virome defined? Is having taken nectar from an infected plant, make the plant virus part of the mosquito virome?

Line 442 – The data presented here is far too limited to make this conclusion.

Line485-486 – The data presented here is far too limited to make this conclusion.

Comments on the Quality of English Language

Only minor editing will be needed.

Reviewer 2 Report

Comments and Suggestions for Authors

The manuscript " Characterization of sylvatic mosquito viromes in the Cerrado biome of Minas Gerais, Brazil: Discovery of new viruses and implications for arbovirus transmission" presents the results for the RNA HTS of several mosquitos species from Cerrado of Minas Gerais State of Brazil.

The results of the study are interesting, however, for a virome research it could have a different approach. The number of viral contigs in the study is low. How  could this be explained?  On the other hand, prior to the characterization of individual virus species, sequences should be confirmed by sanger sequencing. Therefore, I do not recommend to use the word characterization in the title and to describe the study performed.    

Reviewer 3 Report

Comments and Suggestions for Authors

The manuscript entitled “Characterization of sylvatic mosquito viromes in the Cerrado biome of Minas Gerais, Brazil: Discovery of new viruses and implications for arbovirus transmission” by Maia LJ et al., performs a comprehensive characterization of viromes of sylvatic mosquitoes collected from various locations within Minas Gerais state, Brazil. After extracting total RNA from different pools of mosquitoes, the authors performed massive sequencing, assembly and bioinformatic analysis that allowed them to obtain 11 almost complete viral genomes, including 7 new viral genomes.

In my view, the manuscript presents an excellent structure and development, with an adequate introduction and good description of methods. In addition, the manuscript collects and analyzes novel viral data, which can help to understand viral diversity which can help develop surveillance and control strategies for emerging infectious diseases. So, it would be very useful for researchers in areas such as biological control of insects, virology, biotechnology and basic science; and also for the scientific community in general. Therefore, it could be Accept in present form.

I only have two suggestions:

The quality of the figures from No. 2 onwards needs to be improved.

LINE 256: one space missing in “isolate MG clusterswith”

Round 2

Reviewer 2 Report

Comments and Suggestions for Authors

The manuscript was reviewed by authors according to the suggestions and comments of reviewers and its quality was significantly improved.

Therefore the revised version can be accepted for publication.